# The Strigolactone Pathway Is a Target for Modifying Crop Shoot Architecture and Yield

**DOI:** 10.3390/biology12010095

**Published:** 2023-01-08

**Authors:** Jack H. Kelly, Matthew R. Tucker, Philip B. Brewer

**Affiliations:** 1Waite Research Institute, School of Agriculture Food & Wine, The University of Adelaide, Adelaide, SA 5064, Australia; 2Australian Research Council Centre of Excellence in Plants for Space, The University of Adelaide, Adelaide, SA 5064, Australia; 3Australian Research Council Training Centre for Future Crops Development, The University of Adelaide, Adelaide, SA 5064, Australia; 4Australian Research Council Centre of Excellence for Plant Success in Nature and Agriculture, The University of Queensland, Brisbane, QLD 4072, Australia

**Keywords:** shoot branching, bud outgrowth, plant architecture, strigolactone, auxin, BRANCHED1, crop tillering, crop yield

## Abstract

**Simple Summary:**

Plants have developed the remarkable ability to sense their environment and modify their growth to suit changing conditions. This ability is integral for their survival and is facilitated by a range of plant hormones. Strigolactones (SLs) are one type of hormone that play an important role in plant growth response, where they are a key regulator of lateral branching. When growing conditions become poor, the production of SL increases, which reduces the number of branches a plant can make. Although this response may increase a plant’s chances of survival in the wild, it can have the unwanted effect of decreasing crop yield as the number of seed heads on a plant becomes reduced. It has been discovered that natural variations in the SL hormone pathway have been responsible for yield increases in staple crop varieties, such as rice and maize. We propose that new knowledge of the SL pathway and its effect on crop development can be applied using new technologies to develop crop lines with varied SL function, which may aid us in improving crop shoot architecture and yield across varying environments.

**Abstract:**

Due to their sessile nature, plants have developed the ability to adapt their architecture in response to their environment. Branching is an integral component of plant architecture, where hormonal signals tightly regulate bud outgrowth. Strigolactones (SLs), being a novel class of phytohormone, are known to play a key role in branching decisions, where they act as a negative regulator of bud outgrowth. They can achieve this by modulating polar auxin transport to interrupt auxin canalisation, and independently of auxin by acting directly within buds by promoting the key branching inhibitor *TEOSINTE BRANCHED1*. Buds will grow out in optimal conditions; however, when conditions are sub-optimal, SL levels increase to restrict branching. This can be a problem in agricultural applications, as reductions in branching can have deleterious effects on crop yield. Variations in promoter elements of key SL-related genes, such as *IDEAL PLANT ARCHITECTURE1*, have been identified to promote a phenotype with enhanced yield performance. In this review we highlight how this knowledge can be applied using new technologies to develop new genetic variants for improving crop shoot architecture and yield.

## 1. Introduction

The developmental blueprint of seed plants includes the phenomenal ability to adapt to their architecture by growing new organs. Branching is an important component of plant development as branches often carry seeds, so the number of branches influences the maximum number of seeds a plant can produce. This is fundamental for reproductive success in natural environments, and an important component of crop yield. Thus, it is important to understand this phenomenon for agricultural productivity. Branches can be replicated almost ad infinitum by bud production and outgrowth. Axillary buds are formed in the axils of leaves. Some buds never cease development and grow into a branch. Others may cease development and enter a metabolically active but non-growing state in certain environmental conditions. These buds may resume development later if required to replace lost branches or when conditions are more optimal, while others may cease development altogether. The ability to perceive the environment and make these branching decisions is made possible by phytohormones that act within a highly complex signalling network, which allows for chemical signals to be perceived and a growth response to be triggered [1]. Strigolactones (SLs) are one of multiple important signals that play a key role in branching regulation, where they act as a decision for a bud to enter a non-growing state. By acting as a negative regulator of branching, they can modulate plant architecture to optimise growth, particularly when conditions become sub-optimal [2]. Findings have shown that SLs can inhibit bud outgrowth by promoting transcriptional inhibitors, and by modulating auxin transport to influence apical dominance [3]. Buds that are responsive to changes to the environment will grow out if there is ample light and nutrients. However, in sub-optimal conditions, these buds will readily respond to increased SL levels and cease growing. This reduces branch number, aiding in the conservation of resources. Although this response has evolved over time to improve survivability in the wild, it can be counterproductive in crop plants where the primary objective is to maximise yield. Strong reductions in branch or tiller number can reduce the number of seed heads produced, limiting the maximum yield potential of a crop. It has been shown that genetic lines with variations in SL pathway function have been naturally selected over time, which has unlocked significant yield benefits, particularly in domesticated maize and rice, by promoting a shoot architecture with improved yield and harvestability [4,5]. This review seeks to specifically summarise how SLs signal buds to cease development, with a view to determining how we can apply this knowledge to improve crop shoot architecture and yield.

## 2. Strigolactone Classification and Synthesis

SLs are a novel class of phytohormones comprised of several structurally diverse molecules that have been identified to regulate numerous aspects of plant function and development. This includes plant stature, inflorescence architecture, shade avoidance, root architecture, senescence and abiotic and biotic stress tolerance. SLs are also exuded from roots to influence soil microbe symbiosis and parasitic weed germination. Strigol was the first SL isolated from root exudates in cotton, where it was identified as a germination stimulant for the parasitic *Striga lutea* [6]. Improvements in stereochemistry along with ongoing technological advancements have since led to the characterisation of over 30 SLs, with some being widely distributed across plant genera and others being specific to the species from which they were isolated [7]. SLs can be defined as canonical or non-canonical subject to key differences in their chemical structure, with canonical SLs consisting of a tricyclic lactone core (ABC ring) that is connected to a butanolide moiety (D ring) via an enol ether bond [8]. Orobanchol is another SL that was later identified as a germination stimulant for the parasite *Orobanche minor*, with subsequent stereochemical analysis identifying an α-orientated C-ring unlike the β-orientated C-ring seen in strigol-type SLs [9,10]. Since this discovery, newly identified canonical SLs have been classified as strigol-like or obobancol-like subject to the orientation of the C-ring. In contrast, non-canonical SLs lack elements of the conventional A-, B- and C-ring structure, but retain the D-ring (hydroxymethylbutenolide), which is essential for SL activity [11].

Early studies identified that SL stimulants are carotenoid derived when analysing carotenoid deficient hosts. Subsequent genetic screening of shoot branching mutants identified β-carotene isomerase and carotenoid cleavage dioxygenases CCD7 and CCD8 as key enzymes that catalyse the synthesis of SL molecules (Figure 1) [12,13,14]. The discovery of these enzymes allowed for the initial stages of SL biosynthesis to be outlined. The all-trans-β-carotene precursor is converted to 9-*cis*-β-carotene by the β-carotene isomerase, which is then cleaved by CCD7 into 9-*cis*-β-apo-10′-carotenal. CCD8 then converts this to carlactone (CL). CL has the A- and D-ring structure and was later identified to be an endogenous precursor for both canonical and non-canonical SLs [15]. CYP711A1, a cytochrome P450 monooxygenase (CYP450), was first identified in Arabidopsis (*Arabidopsis thaliana*) to act downstream of CCD7 and CCD8 in the SL biosynthesis pathway [16]. It was subsequently found that CYP711A1 catalyses three oxidation reactions that convert CL to carlactonoic acid (CLA) and that this conversion is conserved in vascular plants [17,18]. CLA is important as it acts as the precursor for all known SLs, including 5-deoxystrigol (5DS) and 4-deoxyorobanchol (4DO), which are the precursors for strigol and orobanchol-like SLs [19]. Other conversions downstream of CYP711A1, by a range of CYP450s and other enzymes, facilitate the diversity and species specificity of SLs [7]. One of these conversions involves CLA methyltransferase (CLAMT), which converts CLA to methyl carlactonoate (MeCLA) [20]. Another enzyme, known as LATERAL BRACHING OXIDOREDUCTASE (LBO), then catalyses a further conversion of MeCLA to 1′OH-MeCLA, while also demethylating MeCLA to produce CLA [20,21]. This highlights that LBO is likely a key enzyme for SL diversity, although many aspects of its function along with other conversions in the pathway remain poorly understood. While it is known that different SL types have varied bioactivity, the underlying mechanisms have not been discovered [7]. Elucidating these unknown mechanisms in the SL biosynthesis pathway will be important for developing new variants for investigating the influence of this hormone on plant development and response, and the impact of different SLs in the rhizosphere. Additionally, it has also been observed that carboxylesterase enzymes (CXEs) are involved in SL catabolism and sequestration (Figure 1) [22,23]. Context-specific enzyme gene expression and transport of SLs may also be important for function. Rice CYP450s show distinctive expression responses, and PLEIOTROPIC DRUG RESISTANCE 1 (PDR1) has been identified as a polar transporter in petunia that facilitates short-distance SL transport internally in the plant and out into the rhizosphere [24,25]. Rice plants mutated in a specific CYP450 (*Os900*) failed to show root exudation, but retained normal tillering, indicating a possible biosynthesis pathway specific for root exudation, although further analysis of individual biosynthesis genes is required to unravel these effects [26].

## 3. Strigolactone Signalling Mechanism

The α/β-hydrolase DWARF 14 (D14) was initially identified as the receptor for SLs in rice (*Oryza sativa*) tillering mutants and has since been isolated in numerous species including Arabidopsis (AtD14), petunia (*Petunia hybrida*) (DAD2) and barley (*Hordeum vulgare*) (HvD14) (Figure 1) [27,28,29,30]. D14 forms the core of SL signal perception, where it can directly bind SL molecules. The binding of an SL promotes the interaction between D14 and an F-box protein originally identified in Arabidopsis as MORE AXILLARY GROWTH 2 (MAX2), forming a SKP1-CULLIN-F-BOX (SCF) ubiquitin ligase complex [31]. Observations of *max2* mutants showed the same high tillering phenotype as SL biosynthesis mutants but could not be rescued with treatment of synthetic SL (GR24), highlighting that SL signalling is dependent on D14-MAX2 for proteasome-mediated protein degradation [32]. The target proteins were identified as transcriptional repressors (represented by DWARF 53 (D53) in rice) [33,34,35]. After elucidating the function of these proteins, the general mode of action for SL signal transduction could be proposed. SL binds to D14, which then recruits the MAX2 F-box protein and D53 target proteins to form an SCF complex. D53 then undergoes proteasomal degradation, triggering the downstream SL signalling response [36].

The D14 receptor is somewhat unique compared to other hormone receptors, due to its dual function as a receptor and an enzyme. When SL is bound, the ABC-ring is cleaved from the D-ring, releasing the ABC-ring from D14 resulting in the creation of a ‘covalently linked intermediate molecule’ (CLIM) [37]. It was proposed that the creation of CLIM allows for the conformational change of D14, allowing for interaction with key proteins, such as D53 [38]. However, subsequent findings found that the D-ring may be released as a product of the reaction rather than being bound as CLIM, and that conformational changes in the α-helix of the F-box protein determine D14 conformation [39]. While modelling of the SL signal transduction mechanism is still ongoing, these findings propose that upon binding of SL, the conformation of the α-helix in the F-box protein changes the conformation of D14, which determines if the entire SL molecule is bound, or if the D-ring is cleaved at the enol bridge to regulate SL activity. After D14 conformation change it can interact with the SCF complex and recruit target proteins, where degradation and ubiquitination can then occur to trigger a response [38]. These findings underline the unique properties and significance of D14 in regulating SL signal perception, highlighting it as a key component of SL-mediated growth response. D14 seems to have evolved only in seed plants, perhaps from the receptor of the karrikin pathway, with which it retains some cross-functionality [40]. Although SLs can trigger responses in non-seed plants and microorganisms, the SL receptor in other species remains unknown.

## 4. Strigolactone-Mediated Bud Outgrowth

The involvement of SLs in the regulation of shoot architecture has been extensively investigated since the initial discovery of the shoot multiplication signal (SMS) and the subsequent classification of SLs in high-branching mutants [32]. SLs were first identified to inhibit bud outgrowth in experiments including highly branched *ccd8* (SL biosynthesis) and *max2* (SL signalling) mutants, where it was observed that application of GR24 to buds could rescue *ccd8* branching to wildtype (WT) levels, while having no effect on *max2* [32,41]. It is known that bud outgrowth is regulated by a highly complex network of hormonal signals, including auxin, cytokinins (CKs), gibberellins (GAs), abscisic acid (ABA) and sucrose. Additionally, more broadly there are other effects of SLs that could impact on plant growth and crop yield, such as root architecture and soil microbe symbiosis, as reviewed in [42,43]. Auxin is a key growth hormone that is synthesised in shoot tips, where it then moves rootward via the polar auxin transport system (PATS) [44]. Auxin’s involvement in the regulation of bud outgrowth has been extensively investigated since its discovery by Thimann and Skoog, who showed that removal of the shoot apex in broad bean (*Vicia faba*) stimulated outgrowth of axillary buds, and that application of exogenous auxin to decapitated stumps could repress bud outgrowth [45]. Apical dominance is a longstanding model for auxin-mediated bud repression that has continuously evolved over time. It was initially proposed that auxin synthesised in the shoot apex moves downward into buds to inhibit them directly, although this has since been refuted as auxin from the shoot apex does not enter axillary buds in appreciable quantities, suggesting that it regulates outgrowth indirectly [46]. The auxin canalisation model proposes that auxin forms narrow transport streams that connect auxin synthesising tissues (source) to regions where auxin is being depleted (sink) [3]. Polar auxin transport occurs via the PIN-FORMED (PIN) protein efflux carrier proteins, with PIN1 being integral for facilitating downward auxin flow within the stem [47]. As part of a feedback system, auxin can promote expression of *PIN* genes and localise PINs facing the sink within the plasma membrane to alter the sink strength in the stem [3]. By modulating the sink strength canalisation can be promoted or repressed, determining if an axillary bud grows out into a branch. This also outlines the effect of competitive inhibition, where auxin export from a more mature bud can reduce the sink strength and prevent canalisation from younger buds, allowing it to develop into a branch while other buds remain repressed [48]. Although this informs us that canalisation is a necessary condition for bud outgrowth, experiments have shown that initial outgrowth can still occur in pea (*Pisum sativum*) plants treated with auxin transport inhibitors, suggesting that auxin canalisation is more important for ongoing bud outgrowth, rather than initiation [49].

The interaction between auxin and SL was first identified in Arabidopsis SL biosynthesis mutants which showed elevated levels of PIN1 [50]. Subsequent findings also identified that GR24 only inhibited bud outgrowth in the presence of auxin in the main stem [51]. This suggests that SLs play a key role in the auxin canalisation model, where they are transported upward to repress bud outgrowth via modulating PIN1 levels to promote or repress auxin export from axillary buds. While this infers that SL-mediated bud repression is auxin dependant, it has also been identified that SLs can act downstream of auxin signalling to repress bud outgrowth. Experiments conducted in pea found that applying GR24 could inhibit bud outgrowth, even when auxin was depleted in the stem following decapitation [52]. It has also been observed that application of GR24 to shoots treated with the auxin transport inhibitor 1-N-naphthylphthalamic acid (NPA) can still inhibit bud outgrowth, suggesting that SL can repress branching independently of auxin [53].

This is further supported by the identification of the BRANCHED1 (BRC1) transcription factor. *BRC1* expression is highly localised in developing buds and has been observed to arrest their outgrowth, keeping them in a state of dormancy [54]. Like SL mutants, *brc1* mutants exhibit a high branching phenotype which cannot be rescued with GR24, suggesting that BRC1 functions downstream of SL [52]. *BRC1* expression is also reduced in SL mutants and has been observed to be upregulated by GR24 in pea [55]. This highlights that BRC1 acts as an integrator in SL-mediated branching responses, where auxin promotes SL expression, which subsequently promotes *BRC1* expression in buds to inhibit outgrowth (Figure 1). While SLs act to induce *BRC1* expression inside buds, it has been shown that CK acts antagonistically to repress it [55]. In contrast to SLs, auxin is known to downregulate CK levels, which has been shown to subsequently downregulate *BRC1* to promote bud outgrowth in pea [56]. Experiments in Arabidopsis have also shown that CK can regulate lateral auxin transport by promoting PIN3,4,7, suggesting that it can also influence auxin canalisation independently of BRC1 [57]. GA is another positive regulator of growth that has been linked to branching, with observations in rice showing that GAs regulate SL biosynthesis, and in *Rosa* sp. showing that GA biosynthesis is strongly upregulated in buds during outgrowth [58,59]. GA can also function synergistically with CK to negatively regulate *BRC1* and promote bud outgrowth in *Jatropha curcas* [60]. These findings propose that BRC1 is a central regulator of branching that is modulated by the upstream regulation of SL, CK and GA (Figure 1). Experiments in Arabidopsis have shown that ABA levels decrease in correlation with dormancy release, and that expression is upregulated in wildtype plants treated with red/far red light, but not in *brc1* mutants [61]. These results suggest that ABA can also regulate bud outgrowth via downstream repression of BRC1-mediated branching. The involvement and interaction between these hormones highlights that bud outgrowth is regulated via a highly complex signalling network, where multiple hormonal pathways can promote and repress lateral branching by manipulation of auxin transport, or by independently regulating BRC1-mediated branching [62]. This network forms the basis of the second messenger model for apical dominance, which suggests that apically derived auxin interacts with and modulates other key phytohormones to regulate bud outgrowth [63]. While the proposed models for apical dominance and bud repression continue to evolve, SLs play an essential role in the signalling responses that facilitate branching decisions in plants. This highlights the pathway as a promising target for modifying plant architecture and development to improve productivity.

## 5. Strigolactone-Mediated Responses to Sub-Optimal Conditions

Plants are sessile organisms; therefore, the ability to perceive and adapt to their environment is essential for their survival. This is particularly important when growing in sustained sub-optimal conditions, or when a plant is exposed to a sudden onset of stress. The ability to regulate branch number is a key component of a plant’s phenotypic plasticity, as it can modify its shoot architecture to maximise its reproductive success in variable conditions. Soil nutrient content has a significant influence on shoot architecture and can deleteriously influence growth if nutrient content is lacking. SLs have been identified to play an important role in response to nutrient conditions, where analysis of root exudates has shown that SL production is promoted by low phosphate and/or nitrogen levels in numerous plant species, including rice, tomato (*Solanum lycopersicum*), wheat (*Triticum aestivum*) and sorghum (*Sorghum bicolor*) [64]. This coincides with analysis of gene expression in rice which has shown that SL biosynthesis genes are upregulated under low phosphate conditions, which influences a range of traits, including reduced branching in the shoot [65]. This suggests that SL production increases under sub-optimal conditions to reduce branch number, preventing excessive investment in shoot growth when conditions cannot facilitate it (Figure 2).

The link between SL signalling and nutrient response is further highlighted by the more recent discovery of *NITROGEN-MEDIATED TILLER GROWTH RESPONSE 5* (*NGR5*), which was identified in rice as a transcription factor that is highly responsive to nitrate [66]. It has been identified that *NGR5* is an inhibitor of *D14*, and that over-expression of *NGR5* can reduce sensitivity to nitrogen fertiliser, repressing branching inhibition under reduced nitrogen conditions [66]. Analysis of gene expression in *ngr5* rice mutants also shows upregulation of numerous key SL-related genes, including *D14*, *D3* and *TB1* (*TEOSINTE BRANCHED1*) [66]. This suggests *NGR5* acts to modulate tiller responses to nitrogen by repressing *D14* expression, indicating that *NGR5* acts upstream of SL signalling (Figure 1). *SPL14* (*SQUAMOSA PROMOTER BINDING PROTEIN-LIKE-14*) is another known branching-inhibitory gene that was also found to be upregulated in *ngr5* and downregulated in response to nitrogen application, as was observed with *D14* [66]. *SPL14* has been identified as the *IDEAL PLANT ARCHITECTURE1* (*IPA1*)/*WEALTHY FARMER’S PANICLE* (*WFP*) gene, being a semi-dominant quantitative trait locus (QTL), which is known to be a key regulator of plant architecture in rice [67]. *IPA1* is regulated by the microRNA156 (miR156), and it has been shown that a point mutation in the recognition site can perturb miR156-regulated degradation of *IPA1* mRNA, which promotes an ideal plant architecture for rice with reduced shoot branching [67,68]. Importantly, SPL14 has been identified as a key downstream transcription factor targeted by D53 to regulate SL gene expression and signalling response (Figure 1) [68]. This occurs via the binding of D53 to SPL14 to repress its transcriptional activity, and the additional ability for SPL14 to bind to the D53 promoter to form a feedback loop, which upregulates *D53* expression to promote additional branching [68,69].

In addition to nutrient deficiency, SL biosynthesis is known to be upregulated in other sub-optimal conditions, including drought and salinity stress [64]. When growing the *hvdwarf14.d* (*d14*) SL-insensitive barley mutant in drought conditions, Marzec et al. observed that the mutant produced more tillers than wildtype and exhibited accelerated water loss and reduced stomatal response [70]. This aligns with previous findings that showed stress tolerance could be recovered by application of GR24 to drought or salt-stressed *max* mutants in Arabidopsis [71]. ABA function has been extensively investigated, where it is known to be a key signal for abiotic stress response. It has been proposed that crosstalk occurs between the SL and ABA pathways, as SL biosynthesis mutants in tomato and lotus (*Lotus japonicus*) have reduced ABA accumulation [72]. It has also been proposed that miR156 plays a role in facilitating this crosstalk, where the stress-induced upregulation of SL subsequently promotes synthesis of miR156, which increases plant sensitivity to ABA [73]. Like SL, it is also known that ABA can act as a negative regulator of branching by repressing *PIN1* to modulate auxin transport, which may play a role in repressing bud outgrowth by interrupting canalisation [74]. These findings outline the significant role SLs play in facilitating architectural and developmental responses to varying sub-optimal conditions. It is highlighted that there are numerous targets within the SL pathway both upstream and downstream of D14 that may be useful as a target for manipulation to improve growth responses of crop plants and boost productivity.

## 6. Targeting the Strigolactone Pathway for Crop Improvement

For thousands of years, staple crop varieties have been domesticated by selecting lines that have the most optimum architecture for producing higher yields in agricultural systems. Continued advancements in breeding unlocked significant gains during the green revolution in the 1960s, which saw significant improvement in global grain yields of numerous important crops for food security, including rice, wheat and maize [75]. This was achieved by the development of semi-dwarf varieties which greatly improved many important agronomic traits, including lodging resistance and photoassimilate translocation, which resulted in a major leap in crop yields [76]. Although post-green-revolution shoot architecture has boosted productivity, it has also introduced some undesirable characteristics, such as reduced stress response and decreased nitrogen use efficiency, leading to the overuse of nitrogen [75]. This has caused crop tillering to be highly nitrogen dependent, with insufficient nitrogen input being one of the major causes of wheat yield gaps in Australia [77]. Increasing nitrogen usage comes with an environmental cost, with synthetic nitrogen fertiliser production accounting for approximately 2.1% of global emissions, highlighting a potential benefit of decoupling tiller response from nitrogen [78]. While consistent incremental yield improvements have been achieved since the green revolution era, it is widely understood that another significant breakthrough must be made to make the next leap in yield performance required to sustain the global 2050 population projections [77].

It is considered that improving our understanding of the fundamental mechanisms that facilitate plant growth response will be a key factor in unlocking the next major leap in crop yield performance. These mechanisms have naturally evolved over time to maximise reproductive success, with many being highly conserved in plant genera. Unfortunately, many of the growth responses triggered by these mechanisms are often counterproductive in agricultural applications, where the primary objective is to maximise yield. The total yield produced by a crop is determined by key yield components, such as inflorescences (spikes) per hectare, grains per spike and grain weight. Branching is one of the core target traits in agriculture as each branch (tiller) produces a single spike with a finite number of grain-producing florets, making tiller number a key determinant of grain number. Tillering also influences other important characteristics, such as photosynthetic capacity and source-sink relations; therefore, signalling mechanisms naturally attempt to regulate tiller number in response to conditions to optimise growth [79]. An optimum number of tillers is important for crop plants, as excessive tillering can lead to unproductive tillers and reduced photosynthetic efficiency, while low tiller numbers can reduce biomass, grain number and grain filling capacity [80].

Crop tillering is highly sensitive to the environment and can become limited under numerous sub-optimal conditions. This can make crop yields more variable, as there is a greater dependency on optimal growing conditions to realise the maximum potential yield. Because of this, genetic lines weaker in the SL pathway have been naturally selected in breeding programs over time [81]. However, weakening the SL pathway too far can cause unwanted deleterious effects, such as diminished stress response and soil microbe interactions [70,82]. Some of these effects could be mitigated using SL analogues and mimics. This may include the use of SL analogues as plant growth regulators (PGRs) to recover SL function [82]. SL PGRs could also be useful for treating soil to promote parasitic plant germination to reduce seed reservoirs (suicidal germination strategy), and crop variants with altered types of exuded SL could help manage weed infestation [83,84]. Considering that post-green-revolution crop tillering is highly nitrogen dependent, it is considered that sub-optimal conditions, such as insufficient fertiliser input, will result in upregulation of SL, reducing tiller number, and subsequently, overall yield. As it is known that the SL pathway plays a key role in modifying plant architecture in response to the environment, it is proposed that it is an ideal candidate for manipulation to improve crop growth responses.

Quantitative trait engineering utilises newer technologies to target identified genes of interest that are considered candidates for crop improvement. Quantitative trait engineering generates new intermediate (hypomorphic) alleles that can exploit sweet spots in yield [85]. New genes of interest for potential agricultural benefit are often identified in fundamental research. Additionally, the use of Genome-Wide Association Study (GWAS) has also been successfully used in plants, being recently applied in rice to mine new genes associated with effective tiller number [86]. This method has also successfully identified *OsTCP19* as a haplotype allele relating to the *TB1* family in rice, which promotes a reduced tillering phenotype under high nitrogen, which can promote yield by reducing unproductive tillering [5]. Identified genes of interest can then be examined utilising modern direct editing technologies, such as CRISPR-Cas9, which can be used to disrupt and assess their function. In more recent times, it has been shown that the assay for transposase-accessible chromatin, combined with high throughput sequencing (ATAC-seq), can be successfully applied in plants to map open chromatin for *cis*-elements of target genes [87]. This can identify important regulatory elements which can then be targeted with CRISPR-Cas9 to develop new haplotypes with varied gene response. When combining this with phenotypic screening, this can allow for a much more targeted approach for fine-tuning crop traits to maximise productivity. Liu et al. have recently applied this in maize, where they used ATAC-seq and CRISPR-Cas9 to create weak promoter alleles of *CLE* genes from the CLAVATA-WUSCHEL pathway, which successfully improved multiple important yield-related traits [88]. The benefit of this approach is highlighted in numerous studies that have discovered yield improvements stemming from natural variations in the promoters of important genes. This is exemplified in the *ipa1-2D* allele in rice, which carries a tandem repeat in the promoter region, promoting a significantly higher yielding phenotype with moderately reduced tillering and more productive panicles [85].

It has also been shown that the insertion of a transposable element in the promoter region of *TB1* in modern maize enhances its expression, which facilitates the architectural change within the inflorescence that allows for its domestication [4]. Recent observations in SL mutants in maize have also identified that a more ancestral and poorer yielding inflorescence morphology is promoted in the absence of SLs [89]. Association analysis of haplotypes of the *TaD14-4D* gene in wheat show association with effective tiller number and grain weight, with *TaD14-4D-HapIII* undergoing selection in modern wheat breeding due to it exhibiting an architecture with a minor reduction in tillers and higher grain weight and yield [90]. It has also been shown that tillering and yield can be increased in *TaPIN1-RNAi* transgenic wheat plants, suggesting that by reducing *PIN1* expression, and the subsequent effects on the SL pathway, a yield benefit can be achieved [91]. The *NGR5* gene in rice has been observed to carry a natural variation in its promoter region that increases its expression, which significantly boosts tillering and yield under reduced nitrogen conditions [66]. Importantly, many of these alleles that have been discovered to promote architecture with improved yield performance appear to be SL-pathway related. However, there remains a lack of understanding of whether some of these genes affect yield in other ways independent of SL function. These findings further highlight the critical role of the SL pathway in optimising shoot architecture and promote it as a promising target for future crop improvement.

## 7. Conclusions

Branching is a highly complex and sensitive trait that is integral to optimise a plant’s architecture for its environment. SL’s prominent involvement in bud outgrowth inhibition identifies it as an important regulator of branch number, and the observed upregulation of SL-related genes in response to stress highlights its role in controlling branching under sub-optimal conditions. This infers that the SL pathway is a promising target for improving crop growth responses and productivity, which is supported by the discovery of natural alleles with varied SL response that promote enhanced yielding shoot phenotypes in different crop species. Additionally, the development of safe and cost-effective SL analogues may make widespread use of SL-related PGRs more viable for improving crop productivity. We propose that targeting SL-related genes with a quantitative engineering approach will allow for the creation of new germplasms with varied SL responses, which can inform us of how we can decouple tillering from the environment and other traits to improve crop shoot architecture and yield.

## Figures and Tables

**Figure 1 biology-12-00095-f001:**
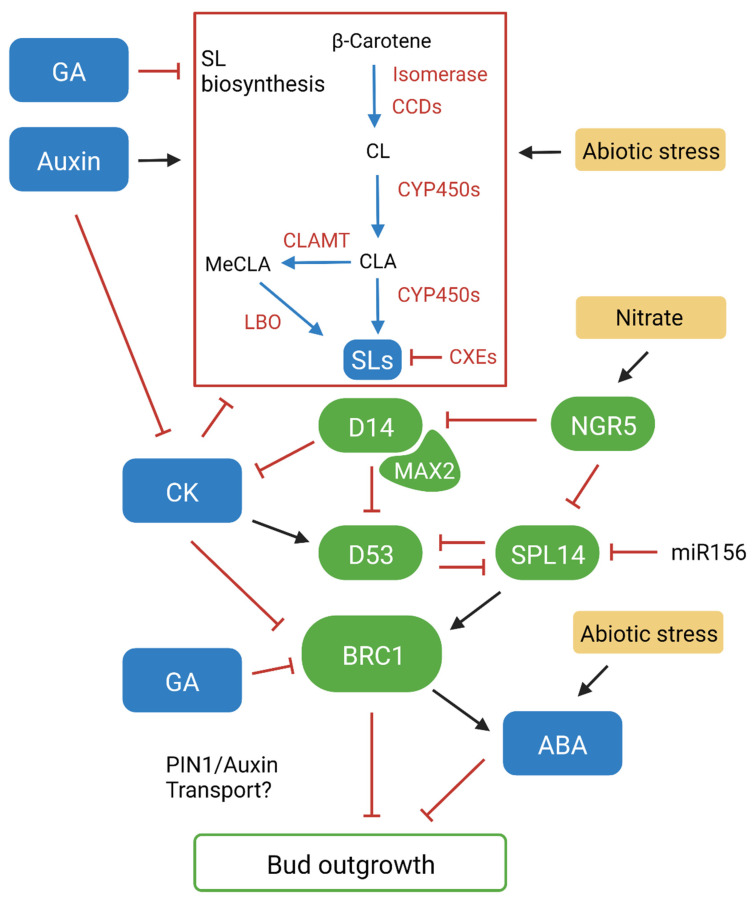
A complex signalling network influences branching decisions by promoting or repressing bud outgrowth. BRANCHED1 (BRC1) plays a central role within this network, acting within buds to repress outgrowth. Auxin and SL act as inducers of *BRC1* while cytokinin (CK) and gibberellin (GA) act to repress it. Abscisic acid (ABA) can also act as a negative regulator of bud outgrowth downstream of BRC1. Blue arrow, conversion; black arrow, promotion effect; red line, inhibition effect; green element, proteins/transcription factors; blue element, phytohormones; yellow element, abiotic condition. Created with BioRender.com.

**Figure 2 biology-12-00095-f002:**
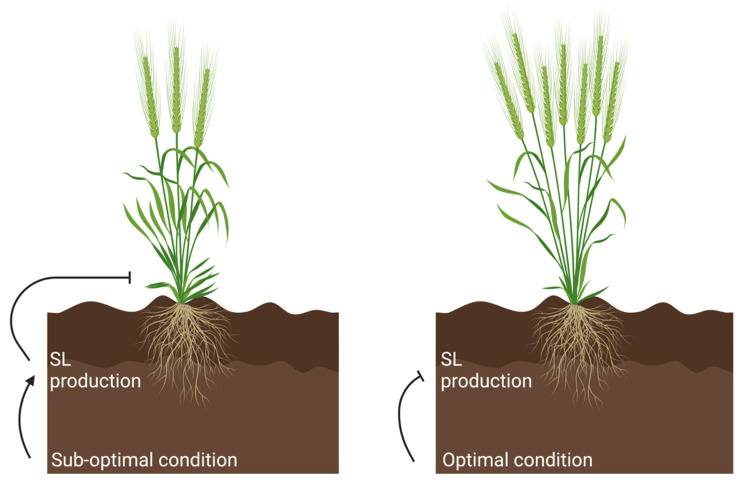
Sub-optimal growth conditions promote the upregulation of SL biosynthesis, which regulates a range of traits, including increased SL-mediated branching inhibition. In more optimal conditions where abiotic stresses (e.g., nutrient and osmotic) are absent, SL biosynthesis is downregulated to promote a shoot architecture with increased branching. Created with BioRender.com.

## Data Availability

Not applicable.

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
