# Peer review of "The Strigolactone Pathway Is a Target for Modifying Crop Shoot Architecture and Yield"

_biology, 2023, doi:10.3390/biology12010095_

Round 1
Reviewer 1 Report
The review "The Strigolactone Pathway is a Target for Modifying Plant Architecture for Crop Yield" focused on a very interesting concept and it’s a new generation plant hormone. Although this study has some positive aspects on IDEAL PLANT ARCHITECTURE 1, some crucial points still need to be revised with SLs regulatory impact and SLs mimic & analogs functions and role in crop yield improvement pathway.
Minor Comments:
Rectify all grammatical errors. If there, spell out all abbreviations in the text if it is the first time mentioned in the text and use SLs abbreviation in Simple Summary. Cross-reference all of the citations in the text with the references in the reference section. Make sure that all references have a corresponding citation within the text and vice versa.
Major Comments:
L27: “…diverse new class..”
L35-37: as you mention “new technologies to develop new genetic material” it's well understood and reported that the application of SLs / GR24/ SLs analogs enhanced crop yield and stimulated the development of the “Suicidal Germination” strategy. Can you define the term “new genetic material”?
L59: Only SLs can make decisions for a bud formation. Are there other hormones or regulators involved?
L74: Use the abbreviation SLs.
L79: “new class of phytohormones”
L130: Figure 1. Include the nitrate structure.
L404: Conclusion: Add future prospects
Author Response
Thanks for the perceptive comments and suggestions. The use of SL-related chemicals in agriculture and the genetic improvement of crop varieties with altered SL exudates for crop soil management strategies is important and innovative. We added text to briefly summarise this at line 374 and cited other reviews on this topic. We did not go into high detail because our focus is on shoot architecture and developmental biology. We slightly altered the text in various places, including the title and abstract, to provide further clarification of this focus.
We also cited 11 additional articles that are either new or we think are pertinent at lines 126, 193, 245, 376, 375, 413, 415 and 419. Some of these articles are reviews with a greater focus on root development and rhizosphere interactions, which we have directed the reader to for more information on this topic.
A modification has also been made to Figure 1 to indicate role of gibberellins in regulation of SL biosynthesis and carboxylesterase enzymes on SL catabolism and sequestration.
Major comments addressed as listed:
L27: “…diverse new class..”
Corrected to “novel class”.
L35-37: as you mention “new technologies to develop new genetic material” it's well understood and reported that the application of SLs / GR24/ SLs analogs enhanced crop yield and stimulated the development of the “Suicidal Germination” strategy. Can you define the term “new genetic material”?
We have corrected to “new genetic variants” relating to developing new lines with varied SL response in the shoot. We mention potential benefit of mimics/analogues using weed management as one example at line 376.
L59: Only SLs can make decisions for a bud formation. Are there other hormones or regulators involved?
We added the statement “one of multiple important signals” at line 59, which we later discuss in the “Strigolactone-Mediated Bud Outgrowth” section.
L74: Use the abbreviation SLs.
We have corrected abbreviations throughout the manuscript including the simple summary.
L79: “new class of phytohormones”.
Corrected to “novel class of phytohormones”.
L130: Figure 1. Include the nitrate structure.
We would prefer to not add this structure. We do not report or discuss chemical structures or their biological relevance in the text. And, if we include one, then the structures of every hormone and signal should be included, which we feel would distract from the main message of the figure.
L404: Conclusion: Add future prospects.
We added some brief extra comments, such as development of more cost-effective SL PGRs at line 439. However, our conclusion already includes the main future prospect relating to shoot architecture, namely the discovery of more genetic variants in the strigolactone pathway with altered tillering responses, either by natural means or by quantitative trait engineering. We agree there are other future prospects, such as for soil management, but these are not part of our main focus on shoot architecture.
Reviewer 2 Report
This review summarised the synthesis and signaling pathways of SLs, and analyzed how strigolactones signal buds to cease development, with a view to determining how we can apply thisknowledge to improve crop plant architecture and yield. As a newly discovered group of phytohormones, SLs did not only play an important role in shoot branching, but also in root architecture to the environment. It is essential to conclude the physiological functions of SLs in the current development. Additionally, the reviewer wish that the function of SLs in root architecture could also be reviewed. The problems in current researches could also be drawn in this paper. In the abstract, the goal of this paper need to supplemented.
Author Response
Thanks for the perceptive comments and suggestions. The impact of SLs on root architecture and how that may relate to crop performance is important. We added text to briefly summarise this at line 82 and 193 and referred the reader to other reviews on this topic. We did not go into high detail because our focus is on shoot architecture. We slightly altered the text in various places, including the title and abstract, to provide further clarification of this focus.
We also cited 11 additional articles that are either new or we think are pertinent at lines 126, 193, 245, 376, 375, 413, 415 and 419. Some of these articles are reviews with a greater focus on root development and rhizosphere interactions, which we have directed the reader to for more information on this topic.
A modification has also been made to Figure 1 to indicate role of gibberellins in regulation of SL biosynthesis and carboxylesterase enzymes on SL catabolism and sequestration.
Reviewer 3 Report
The review is very well written, provides ample details about the strigolactone signalling pathway and is easy to read. The description is comprehensive followed by a sound outlook.
I found only one minor error to correct in line 384: "this allows can allow..."
I recommend acceptance of the review.
Author Response
Thanks for the supportive comments.
We corrected error found in line 384: "this allows can allow...".
We also cited 11 additional articles that are either new or we think are pertinent at lines 126, 193, 245, 376, 375, 413, 415 and 419. Some of these articles are reviews with a greater focus on root development and rhizosphere interactions, which we have directed the reader to for more information on this topic.
A modification has also been made to Figure 1 to indicate role of gibberellins in regulation of SL biosynthesis and carboxylesterase enzymes on SL catabolism and sequestration.